# Influencing Factors of Performance Degradation of Zinc–Air Batteries Exposed to Air

**Yuwei Zhong, Bin Liu, Zequan Zhao, Yuanhao Shen, Xiaorui Liu and Cheng Zhong ***

Key Laboratory of Advanced Ceramics and Machining Technology (Ministry of Education), Tianjin Key Laboratory of Composite and Functional Materials, School of Materials Science and Engineering, Tianjin University, Tianjin 300072, China; zhongyw@tju.edu.cn (Y.Z.); arthurloub@163.com (B.L.); zzq1993220@163.com (Z.Z.); 2019208040@tju.edu.cn (Y.S.); xrliu@tju.edu.cn (X.L.)
* Correspondence: cheng.zhong@tju.edu.cn

**Abstract:** Zinc–air batteries feature high energy density, but they usually suffer from their short storage life after they start working, restricting their commercial applications. In the past, scholars did not reach an agreement on the influencing factors of the performance degradation of zinc–air batteries when exposed to air. Here, a series of comparative experiments were conducted to confirm the changes of the battery during storage after being exposed to air. The morphology and composition of the components of the battery were characterized by scanning electron microscopy (SEM) and X-ray diffraction analyses. SEM images revealed that with the increase of storage days, the corrosion of the zinc anode gradually deepens, but the surface morphology of the air cathode does not change much. The electrolyte of the batteries stored for different periods was examined through inductively coupled plasma spectroscopy and titration. After 20 days of storage, the concentration of $CO_3^{2-}$ reached 2.694 mol $L^{-1}$, which indicates that more than 80% of the $OH^-$ in the electrolyte was consumed. The results show that after being exposed to air, the carbonation of the electrolyte is the main cause of the battery capacity decay.

**Keywords:** zinc–air batteries; storage life; deterioration; corrosion; carbonation

## 1. Introduction

With the development of society and the economy, humans' demand for energy has been growing rapidly [1–6]. Therefore, it is very important to develop clean, efficient and environmentally friendly energy storage devices. In recent years, zinc–air batteries (ZABs) have aroused widespread interest [7]. Due to their unique semi-open structure, zinc–air batteries have a very high theoretical energy density (~1086 Wh $kg^{-1}$), which is about threefold higher than that of lithium-ion batteries [8–11]. Moreover, compared with other metal–air batteries, zinc–air batteries have the advantages of low cost, high safety and low pollution, so it is one of the most promising energy storage devices [12,13].

To promote the commercialization of zinc–air batteries, many research studies have been conducted to improve the energy efficiency and cycle life of zinc–air batteries. However, only a few works focus on the storage life of the zinc–air battery when exposed to air, which is also important for the commercialization of zinc–air batteries. For example, after being exposed to air for 15 days, a flexible zinc–air battery based on the gel KOH-PVA (poly(vinyl alcohol)) electrolyte was found to decrease by more than 90% of its cycling life, which made it unsuitable for the subsequent operation [14]. For the entire future use of commercial secondary zinc–air batteries (e.g., applied in large-scale grid energy storage systems or small electronic devices), it is not always possible to close the air holes to extend its shelf life. Therefore, identifying how to inhibit the self-discharge reaction and extend the storage life of the battery when it starts working is one of the major challenges for the commercialization of zinc–air batteries.

Scholars have put forward various opinions on the causes that lead to the performance degradation of zinc–air batteries. Some scholars believe that zinc corrosion and $H_2$ evolution can affect battery life [15–17]. Henninot et al. [18] stated that gassing may occur before use of the battery, i.e., during its shelf life. This typically occurs after partial discharge. Dongmo et al. [19] analyzed the zinc–oxygen battery during its operation. By means of DEMS, they observed that the ion–current signal of hydrogen is higher during charge than during discharge. Hydrogen is mainly produced by harmful electrochemical reactions at the anode. Li et al. [20] found that the corrosion and dendrite growth of the zinc anode played a major role in battery failure. By replacing the components of the failed battery, they found that the battery using a disassembled Zn anode from the failed battery showed the shortest cycle life. Moreover, the air cathode is another important factor that affects the performance of a zinc–air battery [21]. Min et al. [22] stated that when the cell is recharged at high potentials, carbon corrosion will cause the degradation of the electrode. By contrast, other scholars believe that the effect of $CO_2$ is the most important factor affecting the storage life of zinc–air batteries [23,24]. Jörissen's work [25] showed that the electrolyte is sensitive to $CO_2$ because of the zinc–air battery's semi-open structure, which can react with hydroxide to form carbonates. The working life of the air electrode operating in pure $O_2$ was 10 times that of the electrode in $CO_2$ containing gas. To analyze the impact of the surrounding air composition, Schröder and Krewer [26] developed an isothermal mathematical model of a secondary zinc–air battery with alkaline liquid electrolyte. The results showed that the impact of the surrounding $CO_2$ content is the strongest. Furthermore, a one-dimensional model for both porous electrodes was developed by Stamm et al. [27]. This model was parametrized and validated with a commercial zinc–air coin cell. According to the results, the authors believed that battery lifetime is limited by carbon dioxide absorption into the aqueous alkaline electrolyte. Yang et al. [28] reported that loss of water by evaporation through the air holes was inevitable due to the semi-open structure of the zinc–air battery, and when the water was depleted, the battery ceased to operate.

Although meaningful research studies have been conducted, the inconsistent evaluation standards and benchmarks also seriously hinder the commercialization of zinc–air batteries. For example, Zhang et al. [29] pointed out that the majority of previous studies on rechargeable zinc–air batteries were tested with a small current density and under very shallow cycling depths. Such experimental conditions would not effectively display the actual cycling performances of air cathodes. Hopkins et al. [30] found that some rechargeable zinc–air electrode materials may already be capable of enabling system-level specific energies between 200 and 450 Wh $kg_{sys}^{-1}$. Rechargeable zinc–air batteries will be commercialized if they can achieve such projected specific energy values and demonstrate long cycle lives. Stock et al. [31] analyzed 70 studies of alkaline Zn anodes for application in electrically rechargeable zinc–air batteries reported in the last 20 years based on seven descriptors. They revealed that only a small number of studies provided the data necessary to assess the performance metrics. Inconsistent evaluation standards and benchmarks will mislead scholars' research direction and slow down the commercialization of zinc–air batteries.

Herein, a systematic analysis of each component of the zinc–air battery stored in open air for various days was conducted. Batteries with a sandwich structure were assembled and stored at room temperature to simulate the storage process of secondary zinc–air batteries. To determine the cause of the battery's failure during storage, the anode and cathode of the stored batteries were characterized by X-ray diffraction (XRD) and scanning electron microscopy (SEM). Inductively coupled plasma spectroscopy (ICP) was used to characterize the electrolyte. Moreover, electrochemical impedance spectroscopy (EIS) was used to investigate the overall self-discharge behavior of zinc–air batteries.

## 2. Materials and Methods

### 2.1. Assembly of Zinc–Air Unit Cell

Figure 1 shows the components of the zinc–air battery used in this work. All the batteries were assembled using a commercial zinc–air battery test device (Changzhou

Youteke New Energy Technology Co., Ltd., Changzhou, China) with a width of 7 cm, a height of 6 cm, a reaction area of 3 cm$^2$, and an electrolyte loading of 6 mL. The anode material used was a 0.5 mm thick, 99.9% pure zinc sheet, and the cathode material was a commercial $MnO_2$ air electrode (Changzhou Youteke New Energy Technology Co., Ltd., Changzhou, China). A 6 mol L$^{-1}$ KOH solution was used as the electrolyte. Before battery assembly, the zinc sheets were polished with 400 grit sandpaper and ultrasonicated in deionized water for 2 min. At the same time, the test mold was also thoroughly cleaned with deionized water.

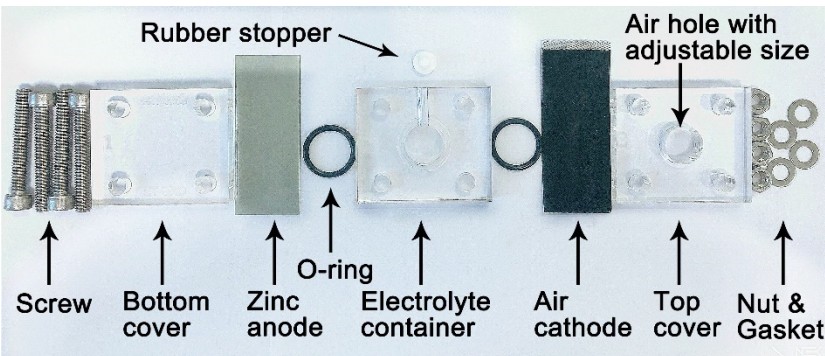

**Figure 1.** Component images of a unit type zinc–air battery.

In the battery assembly process, follow the sequence from left to right in the schematic diagram to assemble the components one by one, and finally tighten them with nuts. Two O-rings are clamped in the electrolyte container's grooves and the plates clamp the cathode and anode. For further research, top covers with different air hole sizes were designed to meet different experimental requirements.

### 2.2. Material Characterization

X-ray diffraction (XRD) patterns of the cathode and anode were obtained on an X-ray diffractometer (D8 Advanced, Bruker) with Cu K$_\alpha$ radiation, in the range of 10° and 90° at a scan rate of 6° min$^{-1}$. The microstructures and composition of the cathode and anode were observed using a field emission scanning electron microscopy (FESEM, JSM-7800F, JEOL, Tokyo, Japan). Inductively coupled plasma (ICP) analysis was performed using an inductively coupled plasma emission spectrometer (Optima 2100DV, PerkinElmer). The content of $CO_3^{2-}$ is determined by titration using saturated $BaCl_2$ solutions.

### 2.3. Electrochemical Characterization

The discharge data (measured at a constant current of 20 mA) of the fabricated batteries stored for different periods were evaluated using a LAND CT2001A eight-channel automatic battery test system. The storage life of the zinc–air batteries was measured at the same discharge condition after different storage times under the packaged condition. The electrochemical impedance spectroscopy (EIS) of the fabricated batteries was obtained using an electrochemical workstation (CHI 660E, Shanghai Chenhua), with an initial potential of 1.2 V.

## 3. Results and Discussion

### 3.1. Whole Battery Analysis

Figure 2a shows the discharge curves of the batteries after storage for 2, 4, 7, 10, and 20 days. It can be seen that with the increase of storage time, the discharge time of the battery is significantly shortened, and the voltage of the battery is reduced. This demonstrates that the zinc–air battery does have a self-discharge problem. After 10 and 20 days of storage, battery capacity has been reduced by about 40% and 80%, respectively.

To further determine the cause of battery performance degradation, electrochemical impedance spectroscopy (EIS) tests were also carried out on the batteries stored for different periods. Figure 2b shows the Nyquist plots of the impedance. The equivalent circuit is in the inset. $R_o$ represents the total ohmic resistances and the contact resistance of the device with an external circuit. $R_{int}$ represents the interfacial resistance between the electrodes and the electrolyte and $R_{ct}$ is the charge transfer resistance [32]. It can be observed that as the storage time increases, the impedance curves of the batteries shift to the right continuously, and the semicircular arcs in the intermediate–low frequency region also expand significantly. Larger X-intercept and semicircular arcs in the intermediate–low frequency region of the plot usually indicate a larger $R_o$ and $R_{ct}$. The $R_o$, $R_{int}$ and $R_{ct}$ values of the batteries were calculated, respectively, based on the equivalent circuit model, and the results are shown in Table 1. The increase of $R_o$ indicates an increase in the resistance of the electrode material and/or electrolyte. Meanwhile, a higher $R_{ct}$ usually means slower kinetics processes in the charge transfer process of the electrode material [33]. The increase in resistance will cause a decrease in the output voltage of the batteries, which is consistent with the discharge curves in Figure 2a.

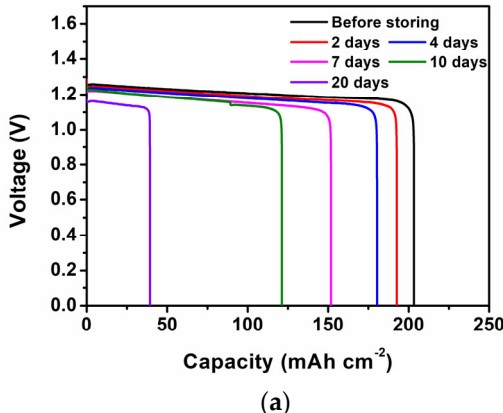 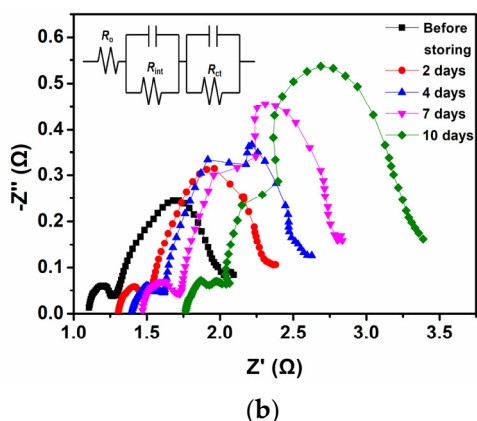

(a) (b)

**Figure 2.** (**a**) Discharge curves of the batteries stored for various days. (**b**) Nyquist plots of the impedance of the batteries before storing and after being stored for various days at a potential of 1.2 V, with an equivalent circuit used in this experiment in the inset.

**Table 1.** Impedance data obtained by fitting the equivalent circuit diagram in the inset of Figure 2b.

| Storage Time | $R_o/\Omega$ | $R_{int}/\Omega$ | $R_{ct}/\Omega$ |
|---|---|---|---|
| before storing | 1.148 | 0.1854 | 0.635 |
| 2 days | 1.327 | 0.2109 | 0.749 |
| 4 days | 1.411 | 0.2178 | 0.876 |
| 7 days | 1.495 | 0.2531 | 1.011 |
| 10 days | 1.788 | 0.2819 | 1.200 |

The key factors resulting in the failure of the zinc–air battery remain unclear and are critical for developing zinc–air batteries with long storage life. Thus, in-depth investigations were carried out to clarify the potential reasons for the failure of the battery. Three batteries were fabricated in the same condition and stored for 10 days. After the storage test, the three batteries were replaced with a zinc anode, an air cathode and an electrolyte, respectively, and then discharged at a constant current of 20 mA. As shown in Figure 3, after 10 days of storage, the discharge capacities of the battery with the replacement of a zinc anode or an air cathode are 94.3 mAh cm$^{-2}$ and 81.0 mAh cm$^{-2}$, respectively, which is even lower than the battery without changing the component (121.3 mAh cm$^{-2}$, green line in Figure 2a). In contrast, after replacing the electrolyte, the discharge capacity of the battery significantly increases by about 76.6 mAh cm$^{-2}$ (197.9 mAh cm$^{-2}$, blue line in Figure 3), which is close to the discharge time of a fresh battery (203.3 mAh cm$^{-2}$, black line in Figure 2a).

This experiment shows that the deterioration of the electrolyte plays an important role in the capacity loss of the battery during storage. To systematically analyze the changes in each component of the zinc–air battery in the storage process, we characterized the anode, cathode, and electrolyte of the stored battery, respectively.

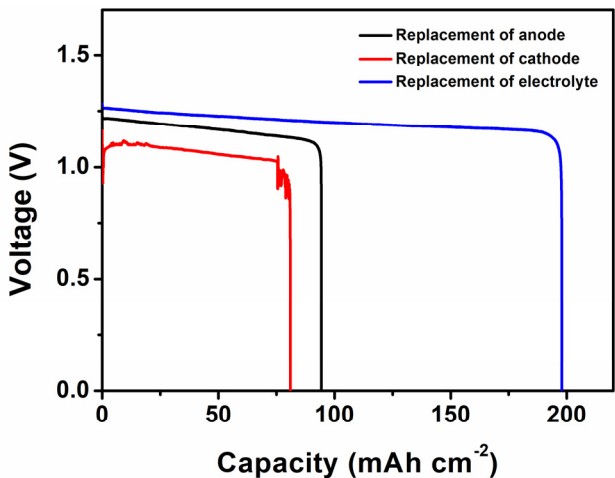

**Figure 3.** Discharge curves of the batteries whose cathode, anode and electrolyte were replaced, respectively, after being stored for 10 days.

### 3.2. Anode Analysis

To determine the morphological changes of the zinc sheets during the storage period, we disassembled the batteries after the storage time of 2, 4, 7, 10, and 20 days and compared the zinc anodes with a fresh one using a scanning electron microscope (SEM). Figure 4a–f shows the SEM morphology of the zinc anode stored for various days. As shown in Figure 4a, the surface of the fresh zinc sheet is smooth and flat. After being stored for 4 days, corrosion pits appear on the entire surface of the zinc sheet (Figure 4c). More obvious holes appear on the zinc sheet after 10 days of storage (Figure 4e). After 20 days' storage, the surface of the zinc sheet has changed from a dense structure to a porous structure (Figure 4f). The results show that in the fabricated battery, the zinc sheets reacted with the electrolyte without an external circuit, causing corrosion of the zinc sheets. This is mainly attributed to the fact that zinc has a more negative reduction potential than hydrogen [34].

The surface morphology of the zinc anode in the battery stored for 20 days was further investigated by a high magnification SEM. As shown in Figure 5a, the formed holes on the zinc sheet have lamellar inner walls, indicating the continuous corrosion of the zinc anode. The sectional view of the zinc anode is shown in Figure 5b. It can be observed that the corrosion of the zinc anode is not only on the surface but continuously extends downward through the formed holes.

As shown in Figures 4 and 5, the corrosion of the zinc anode seems to be continuous during the entire 20-day storage process. However, regarding the discharge of the zinc–air battery without storage, the termination of the discharge would happen in just a few days. The discharge capacity is far lower than that of the theoretical capacity of the zinc anode, which indicates that there is a large amount of zinc not being used. It is widely accepted that in the process of battery discharge, the reaction on the anode consists of two parts [35]:

$$Zn + 4OH^- \rightarrow Zn(OH)_4{}^{2-} + 2e^- \tag{1}$$

$$Zn(OH)_4{}^{2-} \rightarrow ZnO + H_2O + 2OH^- \tag{2}$$

As shown in Equation (2), the produced $Zn(OH)_4{}^{2-}$ would decompose into ZnO and cover the surface of the zinc anode when it reaches its saturation, which will eventually lead to the passivation of the anode and the termination of the discharge. Because the

barrier created by the ZnO is enough to slow the transport of OH$^-$ and Zn(OH)$_4^{2-}$ between the electrode surface and the bulk of the electrolyte, the reaction cannot be sustained [36]. To explore the cause of this phenomenon, X-ray diffraction (XRD) measurements were conducted to characterize the phase and composition of the zinc anode after storage and the result was compared with the zinc anode in a discharged battery.

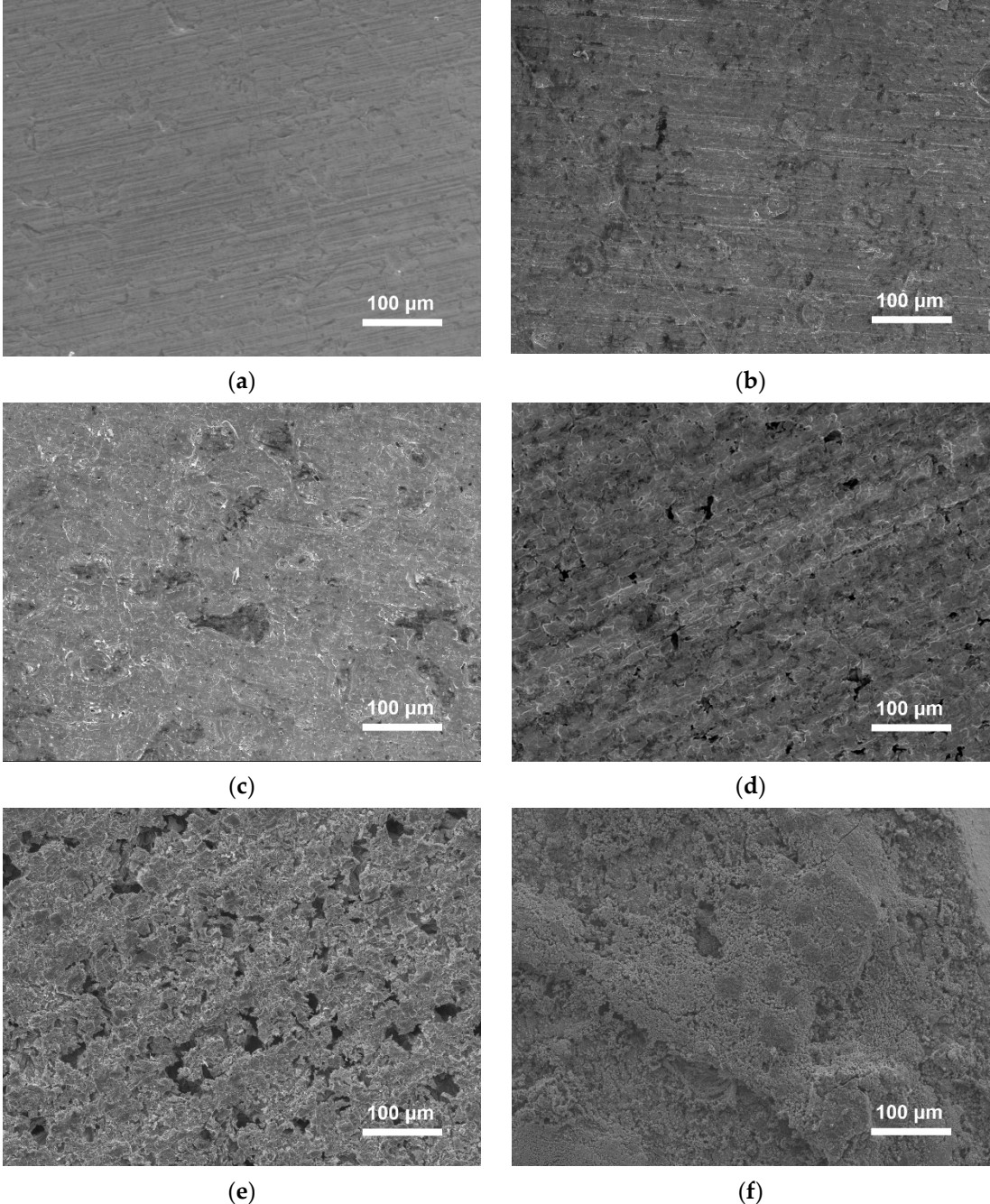

**Figure 4.** Low magnification SEM images of the surface morphology of the (**a**) fresh zinc sheet, and zinc anode in the batteries stored for (**b**) 2 days; (**c**) 4 days; (**d**) 7 days; (**e**) 10 days; (**f**) 20 days.

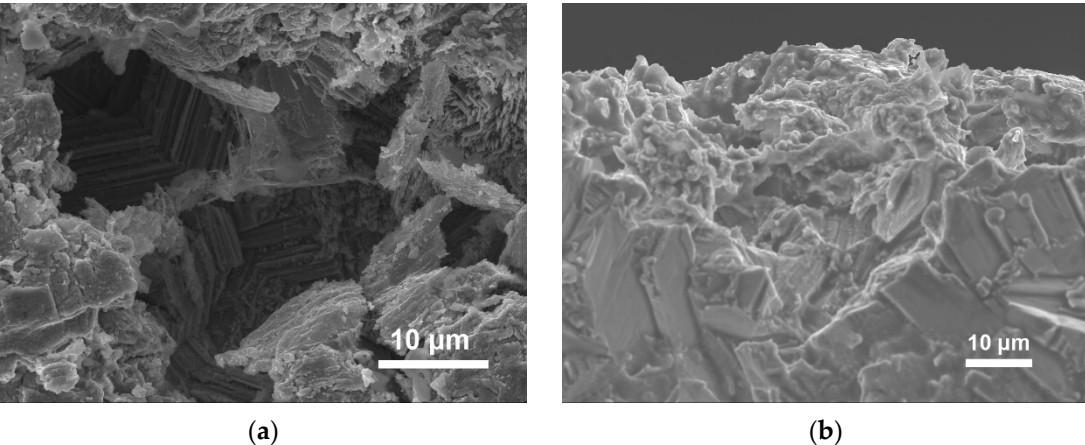

(**a**) (**b**)

**Figure 5.** High magnification SEM Images. (**a**) SEM images of the surface morphology of the zinc anode in the battery stored for 20 days (top view). (**b**) SEM images of the surface morphology of the same zinc anode (sectional view).

Figure 6 shows the XRD patterns of the zinc anodes of a battery discharged without storage and batteries stored for different periods without discharge. It can be observed that with the increase of storage time, the XRD patterns of the stored battery do not show significant change. The peaks at 36.34°, 39.04°, 43.27°, 54.30°, 70.08°, 77.05°, 82.10° and 86.53° are assigned to the (002), (100), (101), (102), (103), (004), (112), (201) reflections of Zn (JCPDS No. 99-0110). Moreover, in the XRD pattern of the zinc anode of the freshly discharged battery, the peaks detected at 31.78°, 34.35°, 36.27°, 47.58°, 56.56°, 62.78°, 67.92°, and 69.09° are attributed to the (100), (002), (101), (102), (110), (103), (112), and (201) reflections of ZnO (JCPDS No. 99-0111). ZnO would be formed on the anode surface after discharge, which is consistent with previous reports [37–39]. However, there are no evident peaks of ZnO on the XRD patterns of zinc anodes in the stored batteries. The possible reason for this could be that the oxidation rate of zinc anode is relatively high during the discharge progress. Thus, excessive $Zn(OH)_4^{2-}$ will be formed on the anode surface to reach its saturation, which leads to the decomposition of $Zn(OH)_4^{2-}$ and the formation of ZnO [40]. In contrast, the oxidation degree of the zinc anodes during storage was significantly reduced. As the corrosion product, the $Zn(OH)_4^{2-}$ cannot reach its solubility limit, thereby ZnO could hardly be generated on the anode surface. Without the protection of the ZnO passivation layer, the fresh zinc surface is continuously exposed to the electrolyte, causing severe corrosion across the anode [41]. The corrosion of the zinc anode will cause the loss of active material. However, according to the experiment mentioned above (Figure 3), the corrosion of the anode is not the main factor causing the decrease of the discharge capacity. This is because there is excess zinc in our battery system. The influence of the zinc anodes is reduced.

### 3.3. Cathode Analysis

To determine whether the composition or morphology of the air cathode changed during storage, XRD and SEM measurements were conducted on the fresh air cathode and air cathode stored in the battery for 20 days. As shown in Figure 7a, the two XRD curves are in good agreement with almost no peak shift, indicating that the composition of the air cathode has hardly changed after 20 days of storage. Moreover, Figure 7b,c shows that after storage, the surface morphology of the air cathode remains unchanged. This also explains why in Figure 3, changing the air cathode cannot improve the discharge capacity of the stored battery.

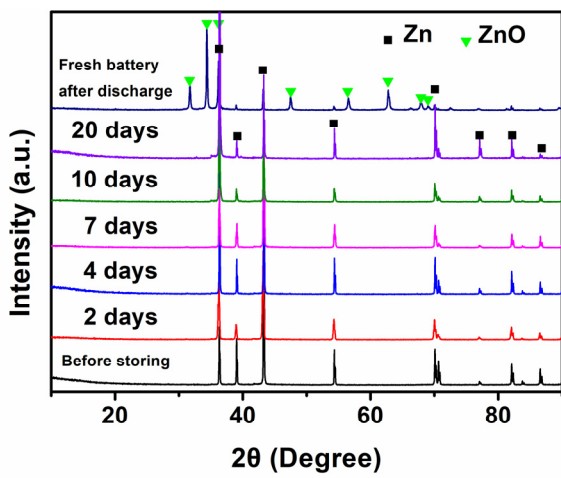

**Figure 6.** XRD patterns of the zinc anode stored for various days.

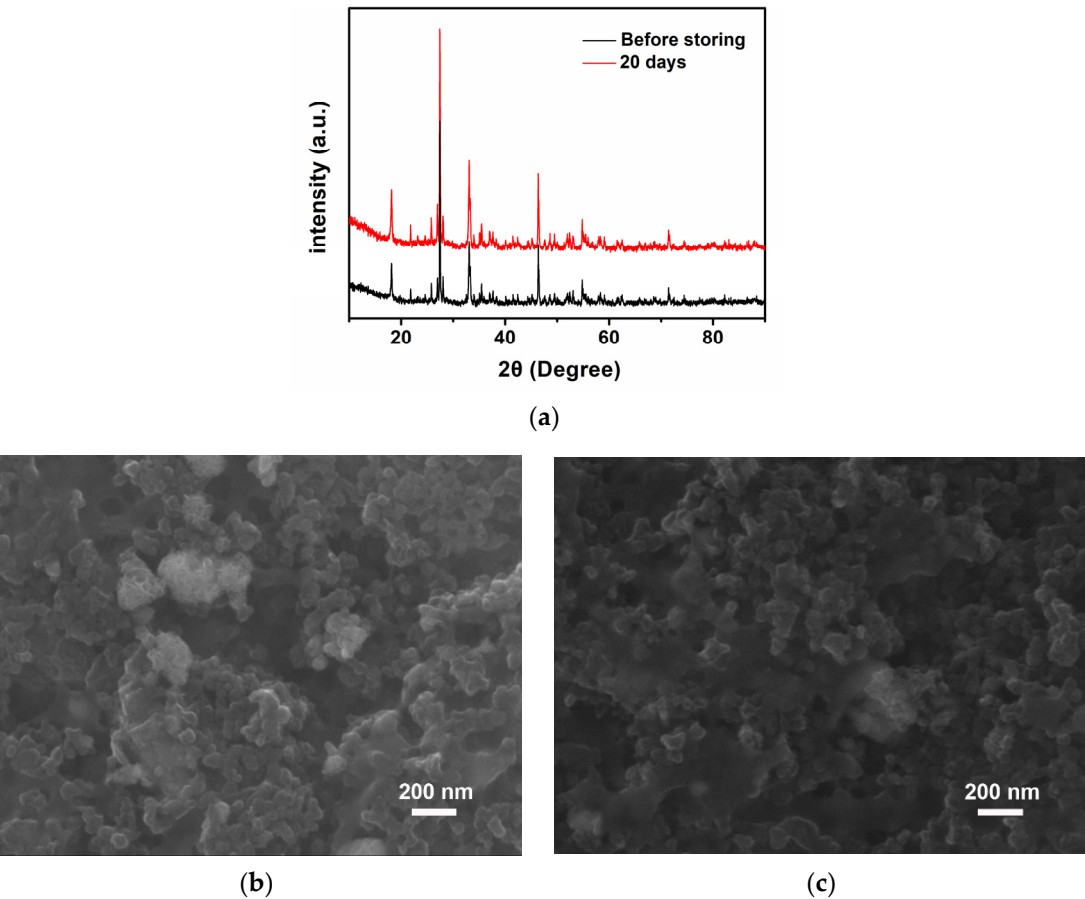

**Figure 7.** (**a**) XRD patterns of the fresh air cathode and air cathode stored for 20 days. (**b**) SEM images of the fresh air cathode and (**c**) air cathode stored for 20 days.

### 3.4. Electrolyte Analysis

The result in Figure 3 shows that the deterioration of the electrolyte is an important factor that affects the discharge capacity of the battery after storage. To explore the changes of the electrolyte during the battery storage, the concentration variation of $CO_3^{2-}$ and $Zn^{2+}$ in the electrolyte was measured. As shown in Figure 8a, with the increase of storage time, the concentration of $Zn^{2+}$ in the electrolyte keeps increasing, reaching about 0.450 mol $L^{-1}$ after 20 days of storage. Moreover, the concentration of $CO_3^{2-}$ reaches 2.694 mol $L^{-1}$

on the 20th day, which means that most of the $OH^-$ in the 6 mol $L^{-1}$ KOH solution was consumed. The result indicates that during the storage of zinc–air batteries, the zinc anode reacts with the electrolyte continuously. Simultaneously, $CO_2$ in the air enters the battery and reacts with the electrolyte, causing serious carbonation. The massive consumption of $OH^-$ leads to a sharp decline in the discharge capacity of the battery. This also explains why replacing the electrolyte of the stored battery can significantly extend the battery discharge capacity (Figure 3).

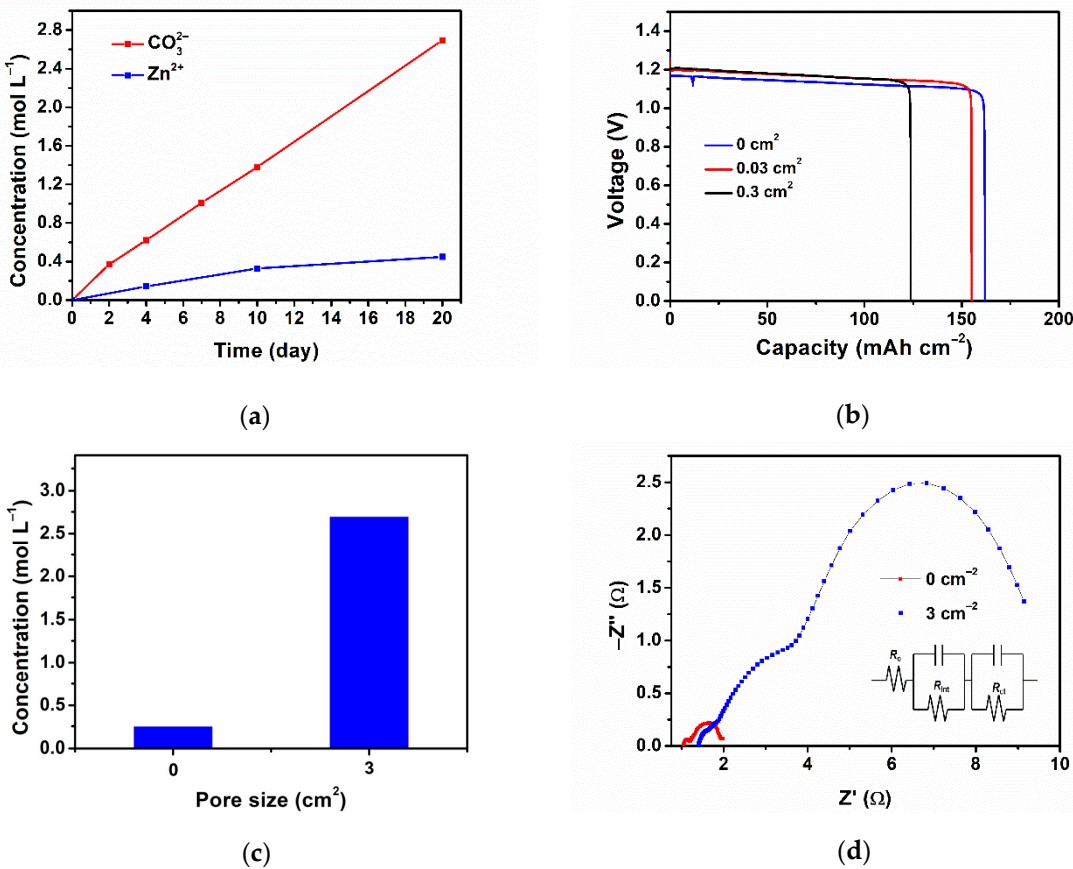

**Figure 8.** (**a**) The concentration variation of $CO_3^{2-}$ and $Zn^{2+}$ in the battery electrolyte as the storage time increased. (**b**) The discharge curves of batteries with different air hole size after being stored under the same conditions for 20 days. (**c**) $CO_3^{2-}$ concentration comparison of the electrolyte stored in the batteries with the air hole sizes of 0 $cm^2$ and 3 $cm^2$ for 20 days. (**d**) Nyquist plots of the batteries with different hole size after being stored for 20 days.

To prove this point, batteries with different air hole sizes (0 $cm^2$, 0.03 $cm^2$ and 0.3 $cm^2$) were designed and performed the same discharge experiment after storing them in the same condition for 20 days (The air hole size of the batteries used in Figures 1–7 is 3 $cm^2$). The results are shown in Figure 8b. It can be observed that the smaller the hole size is, the longer the discharge life of the battery. Moreover, after 20 days of storage, we characterized the electrolyte of the batteries with air hole sizes of 0 $cm^2$ and 3 $cm^2$. The concentration of $CO_3^{2-}$ in the battery with air hole sizes of 0 $cm^2$ is significantly lower than the other one. EIS tests were also carried out on the two batteries. Figure 8d shows the Nyquist plots of the batteries with the air hole sizes of 0 $cm^2$ and 3 $cm^2$ after being stored for 20 days. In the battery with the air hole sizes of 0 $cm^2$, both $R_o$ and $R_{ct}$ are much lower than in the 3 $cm^2$ one. This is mainly due to the fact that $OH^-$ ($20.64 \times 10^{-8}$ $m^2$ $s^{-1}$ $V^{-1}$) has mobility several times higher than $CO_3^{2-}$ ($7.46 \times 10^{-8}$ $m^2$ $s^{-1}$ $V^{-1}$) [42]. The electrolyte with the lower $CO_3^{2-}$ concentration noticeably has smaller ohmic resistances and charge transfer resistance. In summary, according to the results of this experiment, we find that the carbonation of the electrolyte is a key factor leading to the reduction of battery capacity during

storage. This work will help to shape future research activities toward the development of zinc–air batteries with an extended lifetime. More attention should be paid to the carbon dioxide separators or absorbents in order to inhibit the carbonation of electrolyte.

## 4. Conclusions

To verify the causes of performance degradation of zinc–air batteries exposed to air, we used the same mold to perform characterization and electrochemical tests on batteries after storage for different periods. Through the experiment, it was found that the carbonation of the electrolyte is the key factor affecting the storage life of zinc–air batteries. Changing the electrolyte can significantly increase the discharge capacity by about 76.6 mAh cm$^{-2}$. Moreover, after 20 days of storage, more than 80% of the OH$^-$ in the electrolyte was consumed, which corresponds to the results that the battery stored for 20 days lost 81% of its capacity. To further extend the storage life of zinc–air batteries, we suggest that researchers should pay more attention to mitigating the electrolyte's carbonation. In addition, it is suggested to select suitable criteria when evaluating the performance of the batteries. This is due to there being a misunderstanding in predicting the performance of the Zn-based battery tested on lab-scale research in a half-cell configuration. Establishing uniform evaluation standards and benchmarks can help scholars to find the real shortcomings that limit the life of zinc–air batteries.

**Author Contributions:** Conceptualization, C.Z., Y.Z. and B.L.; methodology, Y.Z., C.Z., B.L. and Z.Z.; validation, Y.Z. and Z.Z.; investigation, Y.Z.; resources, C.Z.; data curation, Y.Z.; writing—original draft preparation, Y.Z.; writing—review and editing, Y.Z., B.L., C.Z., Y.S. and X.L.; supervision, C.Z.; project administration, C.Z.; funding acquisition, C.Z. All authors have read and agreed to the published version of the manuscript.

**Funding:** This research received no external funding.

**Acknowledgments:** This research was funded by the National Natural Science Foundation for Excellent Young Scholar (51722403), Tianjin Natural Science Foundation for Distinguished Young Scholar (18JCJQJC46500) and the National Youth Talent Support Program.

**Conflicts of Interest:** The authors declare no conflict of interest.

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
