# Peer review of "Influencing Factors of Performance Degradation of Zinc–Air Batteries Exposed to Air"

_energies, doi:10.3390/en14092607_

Round 1
Reviewer 1 Report
This paper investigates the effects of storage time on zinc-air batteries (ZABs). The ZABs investigated in this work use zinc foil anodes and are stored with open exposure to the surrounding air. The study measures the loss of discharge capacity and changes to the electrochemical impedance spectrum for ZABs stored for a variety of intervals ranging from 2 to 20 days. By replacing individual components and varying the area of the battery air holes, the authors attempt to isolate the dominant sources of degradation.
The core content of the paper is well-structured and clearly presented. The experiments are thorough, and the results could help the zinc-air battery research community to quantify the effects of leaving ZABs open to air for extended periods of time. However, the manuscript makes some claims that warrant further clarification and modification.
First, the authors state repeatedly that "the dominant factors that affect the shelf life of zinc-air batteries remain unclear." This statement is difficult to justify. There are myriad studies going back decades that highlight the role of zinc corrosion and alkaline electrolyte carbonation as the dominant factors affecting zinc-air battery calendar lifetime. While there is widespread agreement in the community that these are the dominant factors affecting calendar life of ZABs, there is less agreement on how to address or quantify these effects. This paper might be better framed as a reference that quantifies how long-term exposure to air affects the performance of lab-scale ZABs, which could be useful to the research community.
Second, the authors claim that they are investigating the shelf life of ZABs, but ZABs are rarely stored on the shelf in the conditions they simulate. Because it is well-known that ZABs deteriorate in air, cells are usually isolated from the air when placed in non-operational conditions. This is achieved by placing a plastic foil over the air holes (primary cells) or purging and isolating the feed gas line (secondary cells).
Third, this study investigates ZABs with pure Zn foil electrodes. While this is common in many lab-scale cells, commercial cells usually feature more advanced Zn electrodes that may be doped or alloyed to limit the corrosion rate. The experiments this study presents do provide valid insights into the performance of research ZAB cells, but they are not necessarily valid for commercial cells.
Although shortcomings exist, the core of the paper is sound and the results are of interest and value to ZAB research. I recommend that the authors conduct a more thorough literature review, examine the points above, and modify their discussion accordingly. For the editor, my recommendation is to reconsider after major revision.
Specific comments follow.
Title: The title is written using an imperative ("Identify"). Consider modifying it to from "Identify the key factors..." to "Identifying the key factors ..."
Keywords: "air hole" is perhaps not the best keyword. Consider something more relevant like "corrosion"
Literature Review: The literature review presented in this manuscript does not sufficiently reflect the vast amount of existing research on this topic. The authors should please conduct a more thorough literature review. As a starting point, please consider the following items:
Electrolyte Carbonation:
(see Section 7) Stamm et al. Modeling nucleation and growth of zinc oxide during discharge of primary zinc-air batteries. Journal of Power Sources 360 (2017) 136-149. DOI: 10.1016/j.jpowsour.2017.05.073
Drillet et al. Influence of CO2 on the stability of bifunctional oxygen electrodes for rechargeable zinc/air batteries and study of different CO2 filter materials Phys. Chem. Chem. Phys., 2001, 3, 368È371. DOI: 10.1039/b005523i
Goh et al. A Near-Neutral Chloride Electrolyte for Electrically Rechargeable Zinc-Air Batteries Journal of The Electrochemical Society, 161 (14) A2080-A2086 (2014). DOI: 10.1149/2.0311414jes
Schröder, D.; Krewer, U. Model based quantification of air-composition impact on secondary zinc air batteries. Electrochim. Acta 2014, 117, 541–553. DOI: 10.1016/j.electacta.2013.11.116.
Corrosion and H_2 Evolution:
M.G. Perez, M.J. O’Keefe, T. O’Keefe, D. Ludlow, Chemical and morphological analyses of zinc powders for alkaline batteries, J. Appl. Electrochem. 37 (2007) 225–231, DOI: 10.1007/s10800-006-9239-3.
Dongmo, et al. Implications of Testing a Zinc − Oxygen Battery with Zinc Foil Anode Revealed by Operando Gas Analysis ACS Omega 2020, 5, 1, 626–633. DOI: 10.1021/acsomega.9b03224
- Henninot, Y. Strauven, ALLOYED ZINC POWDERS WITH PIERCED PARTICLES FOR ALKALINE BATTERIES, 2012, US8142540B2.
Standardization and Benchmarking:
- Stock, S. Dongmo, J. Janek, D. Schr¨oder, Benchmarking anode concepts: the future of electrically rechargeable zinc–air batteries, ACS Energy Lett 4 (2019) 1287–1300. DOI: 10.1021/acsenergylett.9b00510.
B.J. Hopkins, C.N. Chervin, J.W. Long, D.R. Rolison, J.F. Parker, Projecting the specific energy of rechargeable zinc–air batteries, ACS Energy Lett (2020) 3405–3408. DOI: 10.1021/acsenergylett.0c01994.
Citations style: For citations in the text, please add a space between the text and the citation, e.g. "text text text [1]." instead of "text text text[1]." Also check the use of a period in the cross-reference to figures in the text, e.g. "Figure. 4" should be "Figure 4"
English quality: There are many grammar mistakes and awkward phrases in the manuscript. Consider engaging an English editor. Please avoid contractions like "didn't". Other examples from the text include:
Page 1 Line 21: "…different days…" would be more clearly stated as "different periods" or "different intervals". Similar issues on Page 2 Line 83, Page 3 Line 130, Page 4 Line 150, etc.
Page 1 Line 30: " human's " should be " humans' " or " mankind's "
Page 1 Line 36: "3 folds" is "threefold"
Page 1 Line 41: "a lot of researches" should be something like "many research studies". Similar issue on Page 2 Line 78.
Page 2 Line 47: "…reducing >>the<< batteries' performance."
Page 2 Line 53: "the flexible ZAB" should be "a flexible ZAB"
Page 2 Line 55: "the subsequent working" should be "the subsequent operation"
Page 2 Line 84: "Batteries with >>a<< sandwich structure…"
Page 4 Line 134: electrochemical impedance spectroscopy is not capitalized.
Page 5 Line 168: "This means the increase of…" should be something like "This causes/leads to/drives/contributes to the increase of…"
Page 5 Line 169: "The R0, Rint, and Rct >> values<< of the batteries…"
Page 5 Line 182: Avoid personal pronouns like "we" unless absolutely necessary.
Page 6 Line 197: "…during the storage >>period<<"
Page 11 Line 337: "This >>is<< mainly due to >>the fact that<< OH^- has mobility…"
Page 11 Line 359: "scientist" should be plural "scientists" or "researchers"
Figures:
On axes, units are sometimes given in parentheses and sometimes after a slash. Please check which is preferred by the journal and be consistent. If a forward slash is used, please add space between the variable and the unit, e.g. in Figure 8(d) "Z'/Omega" should be "Z' / Omega"
Figure 2(a): Please report the x-axis in mAh*cm^{-2} to facilitate comparison with other studies. The same comment applies to Figures 3 & 8(b) as well as the discharge times reported on page 5 and throughout the text.
Figure 2(b): Why wasn't impedance analysis done on the cell that was stored for 20 days?
Figure 8(b-d): "Pore size" in % is a confusing and ambiguous label for the x-axis. Consider changing to something more specific like "Air hole area" with values 0 cm^2 and 3 cm2.
General Comments to the Text:
Page 1 Line 22: Please subscript the "3" in CO_3^{2-}. Similar issue on Page 3 Line 126.
Page 1 Line 23: Please superscript the "–" in OH^-.
Page 1 Line 36: Please state the theoretical energy density.
Page 2 Line 45: "self-reactions" do the authors mean "self-discharge reactions"?
Page 2 Line 49: What defines the start of the shelf-life period? The end of the manufacturing process?
Page 2 Line 61: "…degradation of the zinc…" Degradation in what way? Corrosion? Something else? Please be more specific.
Page 2 Line 72: The authors should please double check the use of the term "hydroxyl ion". "Hydroxide" is usually the preferred nomenclature.
Page 2 Line 85: The authors state that storing batteries at room temperature simulates the storage of zinc-air batteries. This is not how most zinc-air batteries are stored. When ZABs are stored over extended periods, they are isolated from air to limit electrolyte carbonation and water loss.
Page 8 Line 246: "As a result, the zinc under the passivated surface would be isolated from the electrolyte and not be discharged." This is not necessarily true. Type I ZnO has a porous structure which allows the Zn metal to still contact the electrolyte. Passivation occurs during operation because the barrier created by the ZnO barrier is enough to slow the transport of OH^- and Zn(OH)_4{2-} between the electrode surface and the bulk electrolyte such that the reaction cannot be sustained. However, Zn metal still shares an interface with the electrolyte. Irreversible type II ZnO forms a dense layer over the metal surface and isolates the metal from the electrolyte.
Page 11 Line 338: Consider listing the limiting ionic conductivities and diffusion coefficients of OH^- and CO_3{2-}.
Page 11 Line 352: Consider re-stating what "the replacement experiment" is to help readers.
Reviewer 2 Report
This work "Identify the key factors affecting the shelf-life of zinc-air batteries" shows comparative study for the zinc-battery aging during the storage in terms of the Zn anode and electrolyte, and research for the factors affecting Zn-air battery shelf-life is critical to further improve the electrochemical performance. This is a very nice piece of work, and thus I suggest it to be accepted. And here, I do have one question, for the figure 8b, the 0% air pore size has a lower discharge voltage compared to those of 1% and 10% although its discharge capacity is the largest, why? Does that mean more side reaction (Zn anode, electrolyte) even in the 0% pore size Zn-air cell?
Reviewer 3 Report
The presented manuscript addresses the shelf-life of zinc-air batteries. Such batteries could represent an important step towards a more sustainable energy storage. As outlined, their shelf-life is known to be very limited, however the reasons are not fully understood. Using relatively simple replacement experiments, combined with material characterization techniques, the authors systematically and comprehensively investigate how anode, cathode and electrolyte affect the shelf-life. It was found, that neither anode nor cathode are responsible for the capacity loss during storage and their replacement in a stored cell did not yield improvements. Contrarily, the electrolyte was found to heavily influence shelf life and an electrolyte replacement of a stored cell allows to access the entire capacity again. Further investigations showed that an accumulation of CO32- in the electrolyte took place, indicating the consumption of the necessary OH- species through a reaction with CO2 from the atmosphere. This was verified by applying cathode covers with different hole sizes and hence different area of air contact. The applied characterization methods seem to be well-performed and are presented in a scientific manner. The results might be of interest for the battery community. However, before publication, the following issues should be taken into account:
- On line 70, the authors cite work by Jörissen, which previously showed the influence of the electrolyte and its air sensitivity. Please discuss your own results also in relation to the literature. Which new findings did you achieve in comparison to the cited reference?
- Several graphs are shown throughout the manuscript, containing plots of voltage vs. time. My suggestion would be to plot voltage vs. capacity instead.
- Figure 2b, the impedance data looks much more sophisticated than the used 2RC fitting model. Please explain in the text why you chose this simplified equivalent circuit and provide an example fit vs. data and the R value of the fit.
- For all SEM images (Figure 4, 5) please include scale bars.
- For the influence of the anode, the authors discuss that anode corrosion is not detrimental since there is excess zinc in the battery (l. 283). In practical battery systems however, any excess should be kept at an absolute minimum in order to guarantee high specific energies and low material cost. Can the authors comment on the influence of the anode in the “zero-excess” Zn case?
- Figure 7a, can the reflections of the XRD pattern be assigned to specific components of the cathode?
Round 2
Reviewer 1 Report
The revision provided by the authors adequately addresses the comments raised during the review.
There are some minor capitalization (e.g. Ref 28 zn-air) and special character (e.g. Ref Schro"der) typos in the References list, which should be corrected during the typesetting / proof-checking.
Author Response
The revision provided by the authors adequately addresses the comments raised during the review.
Point 1: There are some minor capitalization (e.g. Ref 28 zn-air) and special character (e.g. Ref Schro"der) typos in the References list, which should be corrected during the typesetting / proof-checking.
Response 1: Thanks for the reviewer’s careful reading. All the typos in the References list have been revised to the correct form.
Specific changes:
(1) In Ref 19, “Schro"der, D.” has been revised to “Schröder, D.”.
(2) In Ref 28, “zn–air secondary batteries” has been revised to “Zn–air secondary batteries”.
(3) In Ref 40, “zn-air batteries” has been revised to “Zn-air batteries”.
